# Morphogenetic degeneracies in the actomyosin cortex

**Sundar Ram Naganathan[1][†][*], Sebastian Fürthauer[2,9], Josana Rodriguez[3,4], Bruno Thomas Fievet[4], Frank Jülicher[2], Julie Ahringer[4], Carlo Vittorio Cannistraci[5,6], Stephan W Grill[1,5][*]**

[1]Max Planck Institute of Molecular Cell Biology and Genetics, Dresden, Germany; [2]Max Planck Institute for the Physics of Complex Systems, Dresden, Germany; [3]Institute for Cell and Molecular Biosciences, Newcastle University, Newcastle, United Kingdom; [4]Wellcome Trust/Cancer Research UK Gurdon Institute, Cambridge, United Kingdom; [5]BIOTEC, Technische Universität Dresden, Dresden, Germany; [6]Brain Bio-Inspired Computing (BBC) Lab, IRCCS Centro Neurolesi "Bonino Pulejo", Messina, Italy; [9]Center for Computational Biology, Flatiron Institute, New York, United States

**\*For correspondence:**
sundar.naganathan@epfl.ch (SRN);
stephan.grill@tu-dresden.de
(SWG)

**Present address:** [†]Institute of Bioengineering, Ecole Polytechnique Federale de Lausanne, Lausanne, Switzerland

**Abstract** One of the great challenges in biology is to understand the mechanisms by which morphogenetic processes arise from molecular activities. We investigated this problem in the context of actomyosin-based cortical flow in *C. elegans* zygotes, where large-scale flows emerge from the collective action of actomyosin filaments and actin binding proteins (ABPs). Large-scale flow dynamics can be captured by active gel theory by considering force balances and conservation laws in the actomyosin cortex. However, which molecular activities contribute to flow dynamics and large-scale physical properties such as viscosity and active torque is largely unknown. By performing a candidate RNAi screen of ABPs and actomyosin regulators we demonstrate that perturbing distinct molecular processes can lead to similar flow phenotypes. This is indicative for a 'morphogenetic degeneracy' where multiple molecular processes contribute to the same large-scale physical property. We speculate that morphogenetic degeneracies contribute to the robustness of bulk biological matter in development.
DOI: https://doi.org/10.7554/eLife.37677.001

## Introduction

Cell and tissue-scale morphogenetic processes are driven by well-orchestrated molecular activities and signalling pathways. Over the last three decades, mutational and RNAi-based genetic screens have provided a catalogue of proteins that are involved in various developmental processes (*Perrimon et al., 2010*; *Mohr and Perrimon, 2012*). But we are still a long way from understanding the mechanisms by which molecular-scale activities drive large-scale events such as actomyosin-driven flows, cell division and migration. To advance this problem, we reasoned that one way to connect molecular activities to large-scale functions is to assess phenotypic consequences of individual gene inhibitions at larger scales, and analyse these in the context of a physical theory. This would allow us to investigate which effective material properties are changed by which molecular perturbation, an important step forward that allows understanding of developmental processes across scales.

We pursued this approach in the context of actomyosin-based cortical flow during polarity establishment in *C. elegans* zygotes. Cortical flow is vital for cell polarization, migration (*Bray and White, 1988*; *Callan-Jones et al., 2016*; *Ruprecht et al., 2015*; *Maiuri et al., 2015*; *Liu et al., 2015*) as well as for formation of contractile ring during cell division (*Turlier et al., 2014*; *Reymann et al., 2016*;

*Salbreux et al., 2009*). In *C. elegans*, cortical flow is driven by a gradient in myosin activity (contractility) (*Mayer et al., 2010*), transporting the anterior polarity proteins, PAR-3, PAR-6 and aPKC to the appropriate region of the cortex to trigger polarization (*Munro et al., 2004*; *Goehring et al., 2011*). Cortical flow dynamics at large scales can be captured using a hydrodynamic theory by considering the actomyosin cortex as a thin film of an active chiral viscous fluid (*Mayer et al., 2010*; *Behrndt et al., 2012*; *Turlier et al., 2014*; *Tjhung et al., 2015*; *Marchetti et al., 2013*; *Fürthauer et al., 2012*; *Fürthauer et al., 2013*). Accordingly, large-scale on-axis (anterior directed) and chiral rotatory flows can be quantitatively described by considering active stresses and active torques (*Mayer et al., 2010*; *Naganathan et al., 2014*). In this approach, two effective bulk material properties are critical in describing large-scale flows: (1) a hydrodynamic length, which determines the distance over which flow generation ranges and which is vital for long-range advective transport of PAR proteins (*Goehring et al., 2011*; *Mayer et al., 2010*) and (2) a chirality index, which quantifies the relative amount by which the cortex is rotated compared to contraction, and which controls embryonic left-right symmetry breaking (*Naganathan et al., 2014*).

At the molecular scale, cortical flow is driven through actin cortex reorganization facilitated by actin binding proteins (ABPs) and actomyosin regulators as well as through force generation by myosin motor proteins. ABPs perform diverse functions such as sequestration of actin monomers and cross-linking, severing, capping and polymerization of actin filaments (*Pollard and Cooper, 1986*; *dos Remedios et al., 2003*; *Winder and Proteins, 2005*). RNAi and mutagenesis screens have given a glimpse of the phenotypes that emerge upon perturbation of ABP function in different processes (*Luo et al., 2013*; *Wiggan et al., 2012*; *Shinomiya, 2012*; *Mendes Pinto et al., 2012*; *Böttcher et al., 2009*; *Girard et al., 2004*). For example, loss of profilin (an ABP that promotes actin polymerization) expression enhances breast cancer cell motility (*Bae et al., 2009*) and loss of actinin (an ABP that crosslinks actin filaments) accelerates cleavage furrow constriction during cytokinesis (*Mukhina et al., 2007*). In the case of cortical flow, however, a comprehensive list of ABPs that regulate flow is not available. Furthermore, it is unknown which molecules regulate the bulk material properties that are critical in describing cortical flow.

Here, we performed a candidate based RNAi screen of 33 ABPs and included 11 actomyosin regulators that were revealed to have a significant impact on flow dynamics, as previously identified in an RNAi-based suppressor screen (*Fievet et al., 2013*). We quantified the impact of RNAi on each of these genes onto large-scale flow dynamics, and asked which molecular activities affect which material properties of the cortex. We identify several proteins to have a significant impact on cortical properties, and we demonstrate that RNAi of proteins with distinct molecular activities leads to similar changes in the quantifications performed. We categorize these proteins into functional groups and uncover degeneracy in the regulation of actomyosin-based cortical flow.

Attempts to link molecular activities of the actomyosin cytoskeleton to large-scale dynamics have been made before. For example, quantifying cellular geometry and using a physical model it was revealed that fish keratocyte shape is predominantly determined by actin network tread milling and membrane tension (*Keren et al., 2008*), but a systematic analysis to relate large-scale behaviours to molecular activities was not pursued in this work. In another study, an RNAi screen was performed in order to investigate cellular morphology and migration in *Drosophila* BG2-cells (*Bakal et al., 2007*). Here, a number of molecular players and signalling pathways were identified, but the connections to large-scale physical activities that determine cellular shape and migration were not characterized. Thus, mechanisms by which molecular activities integrate to determine large-scale physical functions have not yet been revealed. Perhaps the biggest step in this direction is provided by (*Chugh et al., 2017*), where an ABP screen was coupled with a computational model to reveal the principles of regulation of cortex thickness and cortical tension. Our work here provides a complementary approach, and we utilize a coarse-grained physical model to identify molecular activities that contribute to large-scale material properties and the emergent physical behaviours of the actomyosin cortex.

## Results

### Screening approach

We set out to perform a candidate RNAi screen to identify ABPs and actomyosin regulators that affect large-scale properties of cortical flow. We compiled a list of 33 ABPs that had diverse function-alities including nucleation, cross-linking, bundling, severing, capping and sequestering among others (*Supplementary file 1*, see Materials and methods). In addition to ABPs, we included 11 acto-myosin regulators that were identified through a suppressor screen and confirmed to have a signifi-cant impact on actomyosin flow (*Fievet et al., 2013*). To try to obtain a strong loss of function, RNAi of these 44 candidates was performed by feeding L4-staged worms for 40 hrs. Where this resulted in sterile worms or meiotic arrest, we reduced the number of hours of feeding (see Materials and methods, *Figure 1A*, *Supplementary file 1*). RNAi of KLP-1 (a kinesin-like motor protein that is expressed only later in development [*Zhou et al., 2001*]) was used as a negative control for wild-type behaviour (*Hall and Hedgecock, 1991*; *Schonegg et al., 2007*). Under each RNAi condition, we quantified actin cortex structure and dynamics followed by a quantitative comparison to a hydro-dynamic theory (see *Figure 1B* for list of quantifications) to determine the resultant large-scale

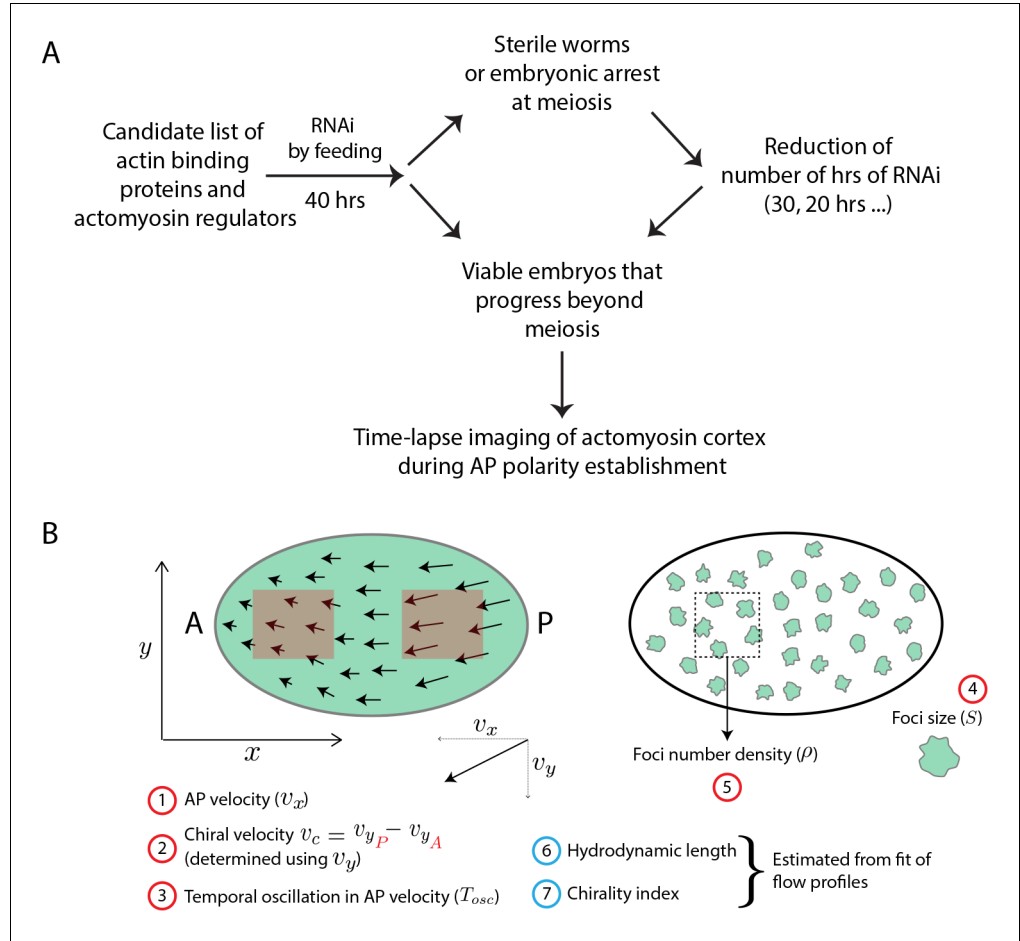

**Figure 1.** Screen strategy. (**A**) Flow chart describes strategy undertaken to perform candidate RNAi screen of actin binding proteins and actomyosin regulators. Time-lapse images were then used to determine flow properties of the cortex. (**B**) Schematic of a 1-cell embryo indicating the quantities measured (numbers in red circle) and estimated (numbers in blue circle). Left - 1-cell embryo, where arrows indicate flow velocities and arrow lengths signify magnitude. A, anterior and P, posterior of the embryo; x-axis is along the long-axis of the embryo and the y-axis orthogonal to it. Red boxes within embryo indicate analysis regions in the posterior and anterior. Right - 1-cell embryo with myosin foci (green blobs).
DOI: https://doi.org/10.7554/eLife.37677.002

material properties of the cortex. In what follows, we first present our different means of quantification of the various RNAi phenotypes and then identify functional groups among the genes knocked down.

## A large fraction of actin binding and regulatory proteins affect cortical flow velocities

To analyse the impact of RNAi knockdown on bulk cortical flow, we measured the cortical flow velocity field by use of particle image velocimetry (PIV, *Raffel et al., 2007*) in embryos containing GFP-tagged non-muscle myosin II (NMY-2). We quantified both the anterior-directed, $v_x$, and the rotational component, $v_y$, of the cortical flow field (*Naganathan et al., 2014*). At each time point during flow, we performed a spatial average in the posterior of the embryo, following which a temporal average over the entire flow period was determined, to obtain an average $\overline{v_x}$. We also determined the difference between spatially averaged $v_y$ in the posterior and anterior region to specify an average chiral counter-rotation flow velocity $\overline{v_c}$. A distribution of $\overline{v_x}$ and $\overline{v_c}$ across all embryos for every RNAi condition was calculated, and the mean values of these distributions were compared to the respective negative controls. We found that 20 of 44 RNAi conditions resulted in a significant change (99% confidence, Wilcoxon rank sum test) in one or both of mean $\overline{v_x}$ and $\overline{v_c}$ compared to the negative control (*Figure 2*, *Figure 2—figure supplement 1*, *Figure 2—figure supplement 2* and see *Videos 1–4* for representative examples). In agreement with previously published results that AP and chiral flow can be modulated independent of each other (*Naganathan et al., 2014*), we observed only a weak correlation between AP and chiral counter-rotation flow velocities (Pearson's linear correlation coefficient, R, 0.04). We observed that on average a reduced anterior-directed or chiral flow could result from either slower flow velocities with the $\overline{v_x}$ or $\overline{v_c}$ distribution respectively closer to zero (*Figure 2B,D* middle, *Video 3*) or from fast flow velocities with a rapid change in direction signified by a broad distribution in flow velocities, for example in *pfn-1 (RNAi)* (*Figure 2B, D* right, *Video 4*). Given that large-scale RNAi screens performed in *C. elegans* 1-cell embryos have failed to identify many of these ABPs (*Sönnichsen et al., 2005*), we show that a rigorous quantification is required to identify subtle yet significant deviations from wild type cortical flow.

## Pulsatility in cortical flow is affected by actin binding and regulatory proteins

In all conditions, we noted that the flow had a pulsatile characteristic (as described in *Nishikawa et al., 2017*) with a periodic temporal increase and decrease in flow velocity (*Video 5*). Pulsatility, emerging through self-organization of the actomyosin network, is an intrinsic feature of active contractile systems and plays an instrumental role in numerous morphogenetic processes (*Gorfinkiel, 2016*; *Munjal et al., 2015*; *Nishikawa et al., 2017*). Identification of molecular components that regulate pulsatility is therefore an important step forward in investigating self-organization of actomyosin networks. We refer to the period of the flow velocity oscillation as $T_{osc}$. To calculate $T_{osc}$, we quantified the temporal change in $\overline{v_x}$ in the posterior of the embryo and performed an autocorrelation of the temporal velocity profile (see Materials and methods for further information). We determined $T_{osc}$ to be $33.3 \pm 4$ s (mean $\pm$ error of mean at 95 % confidence) for the negative control, klp-1 (*RNAi*) condition. We found that 10 RNAi conditions resulted in a significant change in $T_{osc}$ compared to the negative control (*Figure 3*). Importantly, ABPs that regulate actin filament turnover such as F-actin polymerizing proteins (*pfn-1, cyk-1, arx-2*) and F-actin severing (*unc-60, fli-1*) and capping (*cap-1*) proteins have a significant effect on the period of pulsation suggesting that pulsation is regulated by cortex turnover.

## Flow velocities and actomyosin foci structure are correlated

Since network architecture is known to facilitate actomyosin dynamics (*Lieleg et al., 2010*; *Reymann et al., 2012*; *Blanchoin et al., 2014*), we next asked whether a change in flow velocity is associated with an alteration of large-scale actomyosin foci structure (*Munro et al., 2004*; *Munjal et al., 2015*). It was shown previously that a change in actin filament organization highly correlates with a change in myosin organization during cortical flow (*Reymann et al., 2016*). We therefore used non-muscle myosin II as a proxy for cortical structure analysis and quantified myosin foci size, $S$ and foci number density, $\rho$, under each RNAi condition. We defined foci size as the position

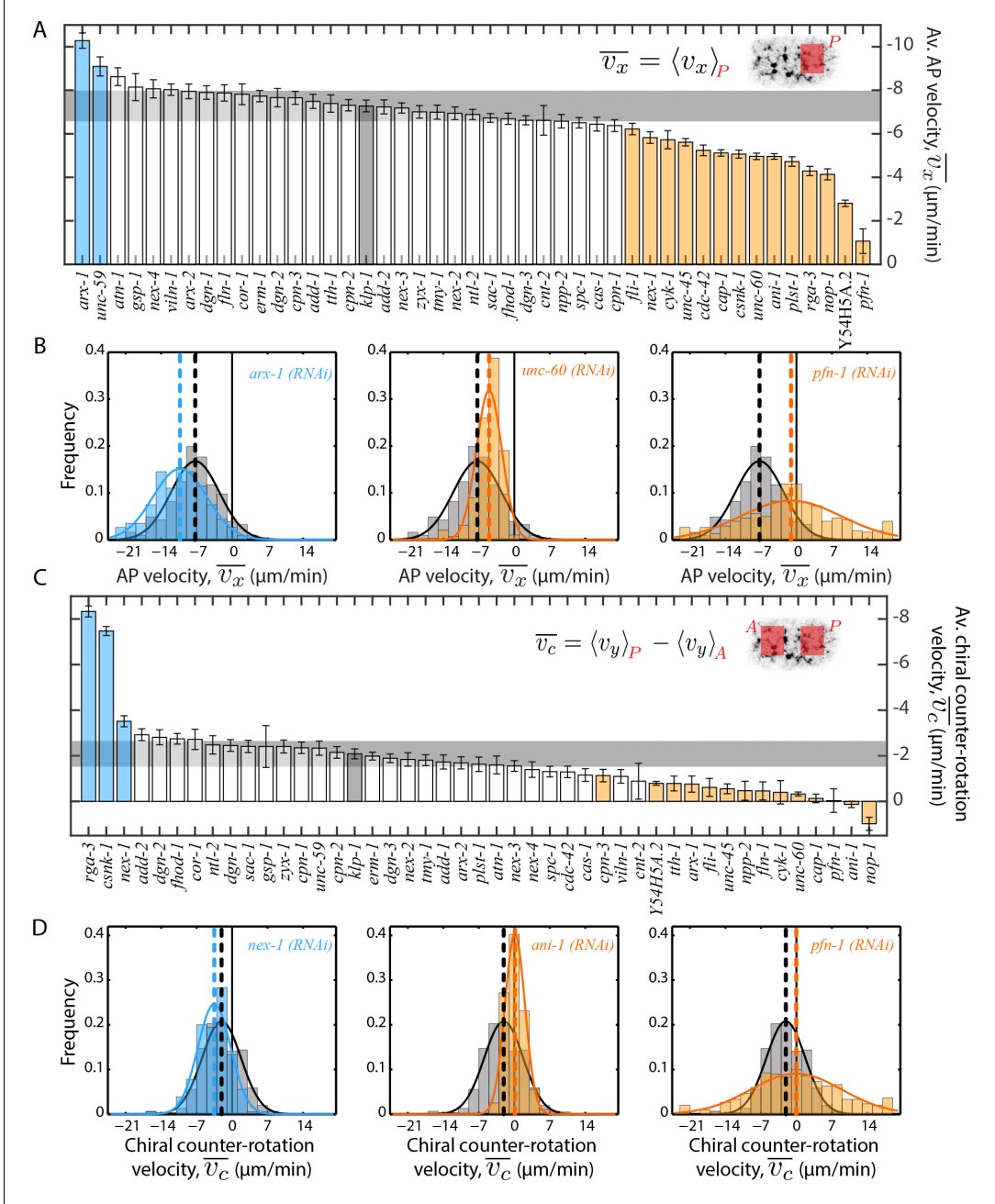

**Figure 2.** Quantification of cortical flow velocities. (**A**), (**C**) Comparison of mean AP velocity $\overline{v_x}$ and mean chiral counter-rotation velocity $\bar{v}_c$ respectively. Shaded areas in the inset represent regions over which the spatial average was performed in each time frame. Error bars, SEM; gray bar, negative control, *klp-1 (RNAi)* condition; gray horizontal bar, error of the mean with 99% confidence for *klp-1 (RNAi)*; cyan, beige bars, significantly different knockdowns with 99% confidence (Wilcoxon rank sum test). (**B**), (**D**) Representative histograms of instantaneous $\overline{v_x}$ and $\overline{v_c}$ respectively. Gray histograms, *klp-1 (RNAi)* condition; dashed lines, mean $\overline{v_x}$ and $\overline{v_c}$ respectively. See *Figure 2—figure supplement 1* and *Figure 2—figure supplement 2* for histograms of significantly different ABPs and *Figure 2—figure supplement 3* for a comparison of mean magnitude velocities. See Supplementary file for number of independent embryo samples in each RNAi condition.

DOI: https://doi.org/10.7554/eLife.37677.003

The following figure supplements are available for figure 2:

**Figure supplement 1.** Comparison of mean AP velocity, $\overline{v_x}$.

DOI: https://doi.org/10.7554/eLife.37677.004

**Figure supplement 2.** Comparison of mean chiral counter-rotation velocity, $\overline{v_c}$.

DOI: https://doi.org/10.7554/eLife.37677.005

*Figure 2 continued on next page*

*Figure 2 continued*

**Figure supplement 3.** Comparison of mean magnitude velocity, $\overline{Mag}$.

DOI: https://doi.org/10.7554/eLife.37677.006

of the first local minimum in the myosin fluorescence intensity autocorrelation function, measured in the anterior region of the embryo (see Materials and methods and *Figure 4A*). To determine foci number density $\rho$, we developed a custom algorithm that identifies foci by analysing variation in myosin fluorescence intensity in the vicinity of identified local fluorescence extrema (see Materials and methods and *Figure 5A*). From these quantifications, we found that 17 of 44 RNAi conditions resulted in a significant change in one or both of myosin foci size (*Figure 4B*, *Figure 4—figure supplement 1*) and foci number density (*Figure 5B*, *Figure 5—figure supplement 1*) when compared to the negative control (*klp-1 (RNAi)*). Interestingly, we observed that $S$ was negatively correlated with $\rho$ (R, −0.84, *Figure 7—figure supplement 2E*), where an increase in foci size corresponded to a decrease in foci number density and vice versa. It remains to be investigated whether there exists a limiting component that leads to this inverse correlation. A comparison of flow velocity with foci size/density revealed that 12 of 17 conditions that affected foci size and/or density also resulted in a significant change in flow velocities. Moreover, $\overline{v_x}$ was negatively correlated with $S$ (R, −0.62, *Figure 7—figure supplement 2B*) and positively correlated with $\rho$ (R, 0.58, *Figure 7—figure supplement 2C*). In other words, bigger foci size (and smaller foci density) is correlated with slower average flow velocities. We conclude that RNAi conditions that affect actomyosin flow velocities tend to also affect myosin foci size and/or density. Importantly, this suggests that actomyosin network structure is strongly linked to the observed dynamics.

## Estimation of large-scale material properties of the cortex

We next sought to identify proteins that affect relevant large-scale material properties of the actomyosin cortical layer. Previous studies have shown that the AP flow velocity ($v_x$) and the $y$-velocity ($v_y$) obey the equations of motion,

$$\partial_x T = \eta \partial_x^2 v_x - \gamma v_x \tag{1}$$

$$\partial_x \tau = \frac{1}{2}\eta \partial_x^2 v_y - \gamma v_y \tag{2}$$

These equations capture many aspects of cortical flow at larger scales through two effective material properties: first, the hydrodynamic length $\lambda = \sqrt{\eta/\gamma}$, which depends on cortex viscosity $\eta$ and $\gamma$ that quantifies the friction with the membrane and/or cytoplasm, and second, the chirality index $c = \tau/T$, where $\tau$ is the active torque density and $T$ is the active tension of the actomyosin cortex (*Fürthauer et al., 2012*; *Fürthauer et al., 2013*; *Mayer et al., 2010*; *Naganathan et al., 2014*). To calculate the material properties, we first determined average myosin distribution along the AP axis (*Figure 6B*, blue markers) during cortical flow, which we then equated to active tension and active torque density. Using *Equations 1 and 2*, we then calculated theoretical AP and chiral flow profiles (*Figure 6B*, *Figure 6—figure supplement 1*, magenta and beige curves respectively) that best matched with experimental flow profiles (*Figure 6B*, *Figure 6—figure supplement 1*, magenta and beige markers respectively). The best fit parameters then yielded the hydrodynamic length and the chirality index. We observed that RNAi of several of the ABPs and actomyosin regulators resulted in a significant change of both the hydrodynamic length and the chirality index as compared to the non-RNAi embryos (*Figure 6A,C*). Thus, material properties of the cortex can be tuned in different ways,

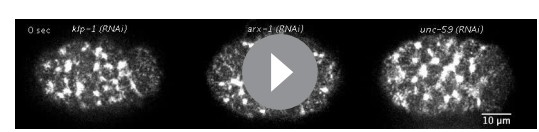

**Video 1.** Phenotype - Fast AP flow: Comparison of cortical flow using TH455 transgenic line (NMY-2::GFP is visualized here) between the negative control, *klp-1 (RNAi)* and *arx-1 (RNAi)* and *unc-59 (RNAi)*, which led to significantly faster anterior-directed flow velocities. Movies were acquired with a 5 s frame interval and are being played 75x faster than real time. Scale bar, 10 μm.

DOI: https://doi.org/10.7554/eLife.37677.007

**Video 2.** Phenotype - Fast chiral flow: Comparison of cortical flow using TH455 transgenic line (NMY-2::GFP is visualized here) between the negative control, *klp-1 (RNAi)* and *nex-1 (RNAi)* and *csnk-1 (RNAi)*, which led to significantly faster chiral flow velocities. Movies were acquired with a 5 s frame interval and are being played 75x faster than real time. Scale bar, 10 μm.
DOI: https://doi.org/10.7554/eLife.37677.008

**Video 3.** Phenotype - Slow AP and chiral flow: Comparison of cortical flow using TH455 transgenic line (NMY-2::GFP is visualized here) between the negative control, *klp-1 (RNAi)* and *cap-1 (RNAi)*, *unc-60 (RNAi)* and *ani-1 (RNAi)*, which led to significantly slower anterior-directed as well as chiral flow velocities. Movies were acquired with a 5 s frame interval and are being played 75x faster than real time. Scale bar, 10 μm.
DOI: https://doi.org/10.7554/eLife.37677.009

by modulating a diverse set of molecular activities. Interestingly, while a large fraction of RNAi knockdowns resulted in a significantly reduced chirality index, three of four knockdown conditions (*rga-3* - also shown in (*Naganathan et al., 2014*), *csnk-1*, *cpn-1*) that resulted in a significantly higher chirality index are known to also impact on myosin activity. This further strengthens the idea that active processes in the actomyosin cytoskeleton generate active torques for chiral symmetry breaking. Notably, 15 of 20 conditions that affected flow velocities and 14 of 17 conditions that affected foci structure resulted in a significant change in the hydrodynamic length. Similarly, 15 of 20 RNAi conditions that affected flow velocities and 13 of 17 conditions that affected foci structure resulted in a significant change in the chirality index. Thus, we conclude that RNAi conditions that affect actomyosin flow velocities, myosin foci size, or myosin foci density tend to also modify the hydrodynamic length and/or the chirality index.

## Dimensional reduction analysis reveals clusters with similar effects on cortical mechanics

RNAi of ABPs and actomyosin regulators resulted in a spectrum of changes in flow velocities, foci structure and material properties of the cortex. Based on the quantifications performed, we next asked whether there exists groups of proteins that have a similar impact on large-scale actomyosin behaviours. To answer this question we used a nonlinear dimensional reduction algorithm to perform a mapping of the dataset to a lower-dimensional space. We then identified clusters of conditions with similar behaviours by manual inspection. We used five parameters for the analysis – foci number density, AP velocity, oscillatory period, hydrodynamic length and chirality index. We computed pairwise distances between the different RNAi conditions using the minimum curvilinear embedding (MCE) approach (*Cannistraci et al., 2010*; *Cannistraci et al., 2013*; *Muscoloni et al., 2017*) (see Materials and methods for more information). Dimensional reduction of this data set (*Figure 7A*) revealed four different groups of proteins with similar effects on cortical mechanics. Cluster zero corresponds to the wild type (Wt, *Figure 7A*, *Figure 7—figure supplement 1*). Manual inspection of the pattern by which genes affected the different parameters (heat map in *Figure 7B*), motivated us to split cluster three into three sub-groups of proteins (clusters 3a-3c). Interestingly, each cluster comprised proteins with different functions suggesting that large-scale actomyosin behaviours can be attained by tuning diverse molecular mechanisms.

We next briefly discuss each of the three clusters. Cluster 1 consisted of two actomyosin regulators, the Rho-1 regulator RGA-3 and casein-kinase 1 (CSNK-1) and two ABPs, an actin-membrane cross linker (annexin, NEX-1) and an inhibitor of myosin ATPase activity (calponin, CPN-1) (red, *Figure 7*). A characteristic feature of this cluster was an increase in the average chirality

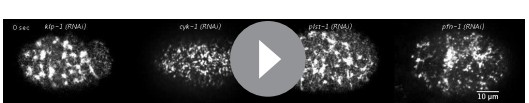

**Video 4.** Phenotype - Short-range flow: Comparison of cortical flow using TH455 transgenic line (NMY-2::GFP is visualized here) between the negative control, *klp-1 (RNAi)* and *cyk-1 (RNAi)*, *plst-1 (RNAi)* and *pfn-1 (RNAi)*, which led to short-range flow. Movies were acquired with a 5 s frame interval and are being played 75x faster than real time. Scale bar, 10 μm.
DOI: https://doi.org/10.7554/eLife.37677.010

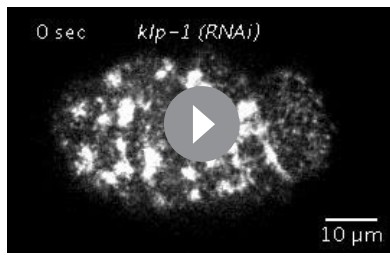

**Video 5.** Pulsatile flow: Cortical flow exhibits pulsatile characteristics in the posterior cortex. Cortical flow acquired using TH455 transgenic line (NMY-2::GFP is visualized here for the negative control, *klp-1 (RNAi)*) with a 5 s frame interval. Movie is being played 25x faster than real time. Scale bar, 10 μm.
DOI: https://doi.org/10.7554/eLife.37677.012

index suggesting that these proteins regulate chiral properties of the actomyosin cortex.

Cluster 2 consisted of an actomyosin regulator (NTL-2, a CCR4-NOT transcription complex subunit) and ABPs that scaffold the cortex (SPC-1, spectrin) and create branched actin networks (ARX-2, a component of the ARP 2/3 complex) (green, *Figure 7*). RNAi of cluster two proteins resulted in an increase in average foci density and a decrease in the period of oscillation, that is faster oscillations. RNAi conditions that resulted in faster oscillations were also found in cluster 3b (see below). Together, these knockdowns enable investigation of mechanisms that lead to emergence of pulsation.

The third cluster consisted of a diverse array of proteins. We split this cluster into three subgroups which displayed similar behaviours in terms of the quantifications performed. Cluster 3a consisted of 6 actomyosin regulators (orange, *Figure 7*) displaying an increase in the hydrodynamic length and a decrease in the chirality index of the cortex. Cluster 3b (pink, *Figure 7*) consisted of one actomyosin regulator (CDC-42, a Rho-GTPase) and 4 ABPs that promote polymerization (profilin, PFN-1; formin, CYK-1), crosslinking (anillin, ANI-1) and bundling of actin filaments (*Ding et al., 2017*) (plastin, PLST-1). RNAi of cluster 3b proteins resulted in a decrease in average foci number

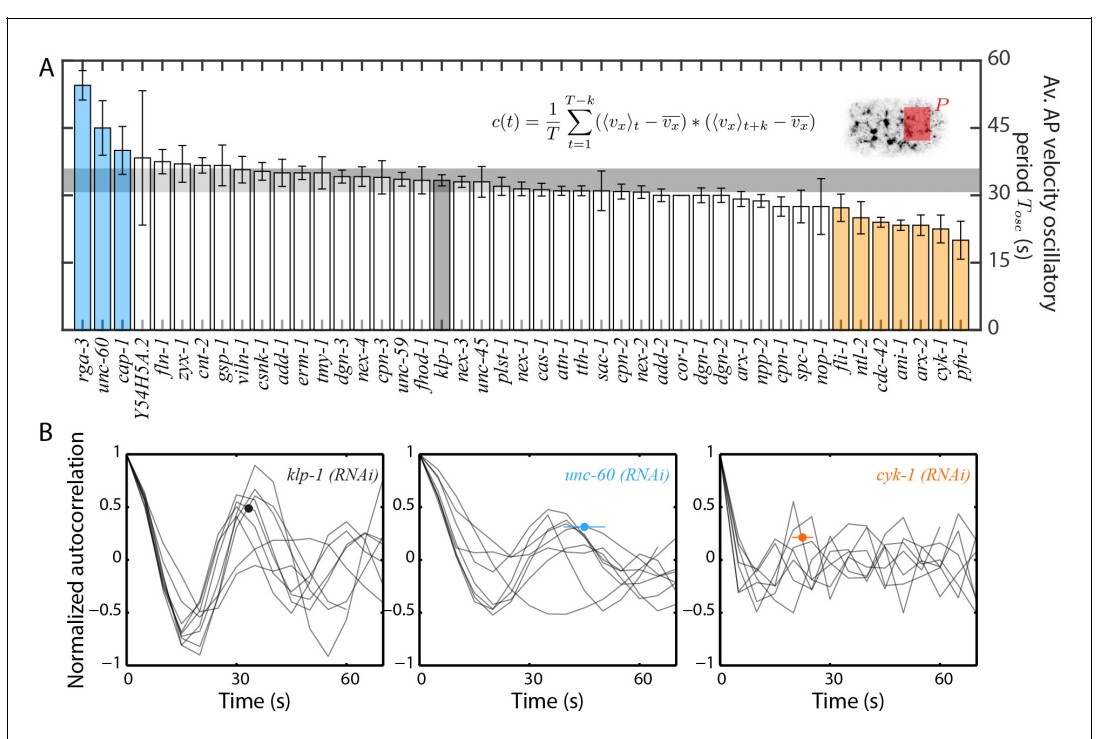

**Figure 3.** Quantification of pulsatile period of cortical flow. (**A**) Comparison of mean period of pulsatile AP velocity, $T_{osc}$, in the posterior of the embryo. Shaded area in the inset represents the region over which the spatial average of velocity was performed in each time frame. The formula for the autocorrelation function is indicated, where $T$ is total analysis time of cortical flow and $k$ is increment (see Materials and methods). Error bars, SEM; gray bar, negative control, *klp-1 (RNAi)* condition; gray horizontal bar, error of the mean with 95% confidence for *klp-1 (RNAi)*; cyan, beige bars, significantly different knockdowns with 95% confidence (Wilcoxon rank sum test). (**B**) Normalized autocorrelation decay curves from individual embryos (thin black lines) for the negative control, *klp-1 (RNAi)*, *cyk-1 (RNAi)* and *unc-60 (RNAi)*. Circular markers represent mean of the periods determined from individual embryos, with error bars representing SEM. See Supplementary file for number of independent embryo samples in each RNAi condition.
DOI: https://doi.org/10.7554/eLife.37677.011

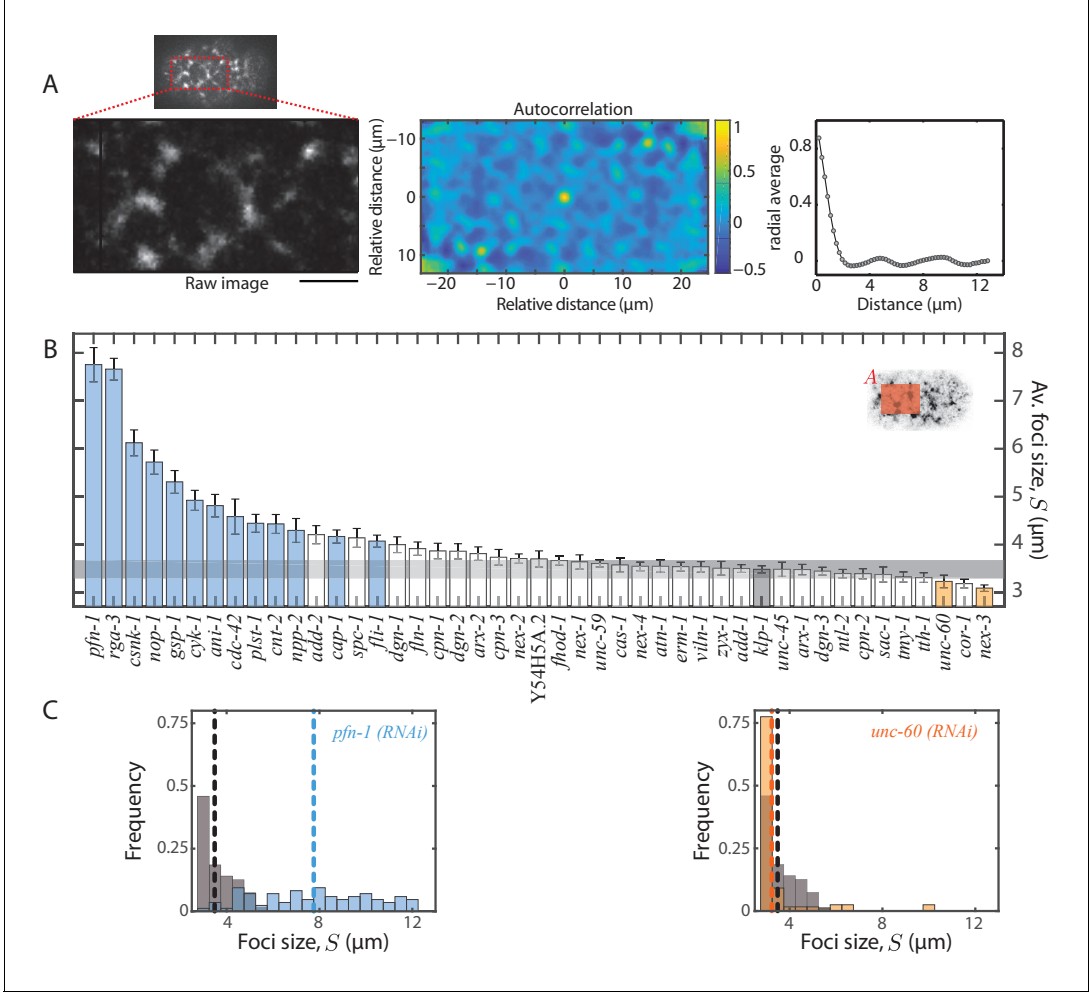

**Figure 4.** Quantification of myosin foci size. (**A**) Representative 2D myosin fluorescence intensity autocorrelation function as a heat map (middle) for the anterior region (left) of a single frame during cortical flow from a *klp-1 (RNAi)* embryo. Scale bar, 5 µm. Right - radial average of the autocorrelation function. (**B**) Comparison of mean foci size, *S*. Error bars, SEM; gray bar, negative control, *klp-1 (RNAi)* condition; gray horizontal bar, error of the mean with 99% confidence for *klp-1 (RNAi)*; cyan, beige bars, significantly different knockdowns with 99% confidence (Wilcoxon rank sum test). Shaded area in the inset represents the region in which the fluorescence intensity autocorrelation was performed in each frame. (**C**) Representative histograms of *S* determined over time during the cortical flow period. Gray histograms, *klp-1 (RNAi)* condition; dashed lines, mean foci size. See *Figure 4—figure supplement 1* for histograms of significantly different ABPs. See Supplementary file for number of independent embryo samples in each RNAi condition.

DOI: https://doi.org/10.7554/eLife.37677.013

The following figure supplement is available for figure 4:

**Figure supplement 1.** Comparison of mean foci size, *S*.
DOI: https://doi.org/10.7554/eLife.37677.014

density, AP velocity, oscillatory flow period and chirality index of the cortex. A lower foci density indicates a less dense actomyosin cortex, hinting at a reduction in effective viscosity of the cortex. Consistent with this possibility, we observed a decrease in the hydrodynamic length of the cortex for RNAi of cluster 3b proteins (*Figure 7B*). Finally, cluster 3c consisted of a myosin chaperone (UNC-45) and two ABPs that cap barbed ends (capping protein, CAP-1) and sever actin filaments (cofilin, UNC-60) (cluster 3c, violet, *Figure 7*). RNAi of cluster 3c proteins resulted in a decrease in average AP velocity, chirality index and an increase in the average period of oscillation (in *cap-1* and *unc-60 (RNAi)* conditions) that is slower oscillations. Furthermore, we observed an increase in foci number density (UNC-60) or foci size (CAP-1), and cluster 3c RNAi conditions (UNC-60 and CAP-1) tend to display a reduced hydrodynamic length. To rationalize the latter observation, we imaged

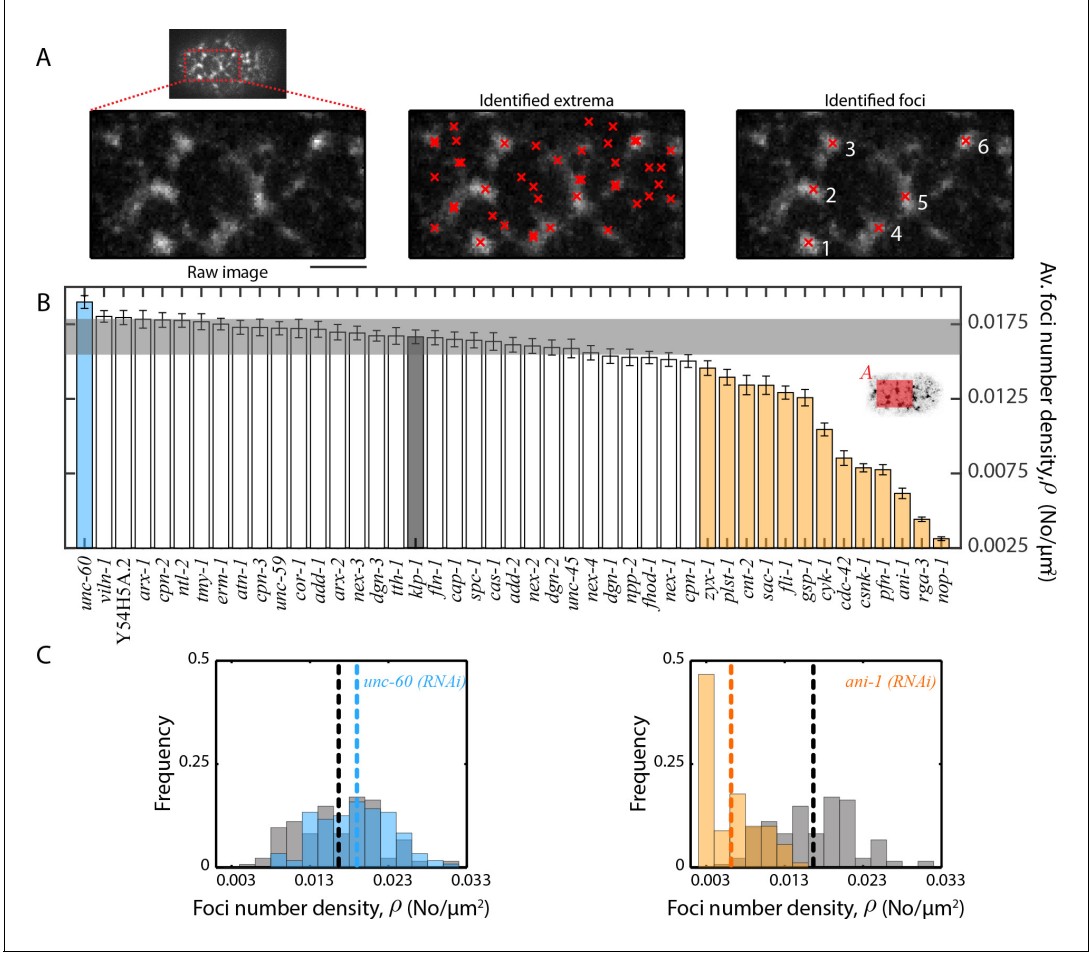

**Figure 5.** Quantification of myosin foci number density. (**A**) Detection of myosin foci (right) by analysis of local changes in fluorescence intensity in the vicinity of identified local extrema (middle) for the anterior region (left) of a single frame during cortical flow from a *klp-1 (RNAi)* embryo. Scale bar, 5 μm. (**B**) Comparison of mean foci number density, $\rho$. Error bars, SEM; gray bar, negative control, *klp-1 (RNAi)* condition; gray horizontal bar, error of the mean with 99% confidence for *klp-1 (RNAi)*; cyan, beige bars, significantly different knockdowns with 99% confidence (Wilcoxon rank sum test). Shaded area in the inset represents the region that was utilized for foci detection in each frame. (**C**) Representative histograms of $\rho$ determined over time during the cortical flow period. Gray histograms, *klp-1 (RNAi)* condition; dashed lines, mean foci number density. See *Figure 5—figure supplement 1* for histograms of significantly different ABPs. See Supplementary file for number of independent embryo samples in each RNAi condition.

DOI: https://doi.org/10.7554/eLife.37677.015

The following figure supplement is available for figure 5:

**Figure supplement 1.** Comparison of mean foci number density, $\rho$.
DOI: https://doi.org/10.7554/eLife.37677.016

cytoplasmic actin using a Life act::mKate transgenic line, and observed an increase in cytoplasmic F-actin density under RNAi of the cluster 3c genes *cap-1* and *unc-60* (*Figure 7—figure supplement 3*). Increasing F-actin density throughout the cytosol is likely to increase cytoplasmic viscosity, which in turn would lead to an increased effective friction and a reduced hydrodynamic length.

Taken together, we identify clusters of ABPs that regulate large-scale material properties of the cortex. A central result from our study is that distinct ABP molecular activities can have a similar impact on large-scale actomyosin cortical behaviours.

## Discussion

Here, we present a systematic approach to connect molecular functions to emergent large-scale physical properties of the actomyosin cortex. We combined RNAi screening with phenotypic characterization in terms of hydrodynamic theory. Our RNAi-based knockdown conditions do not

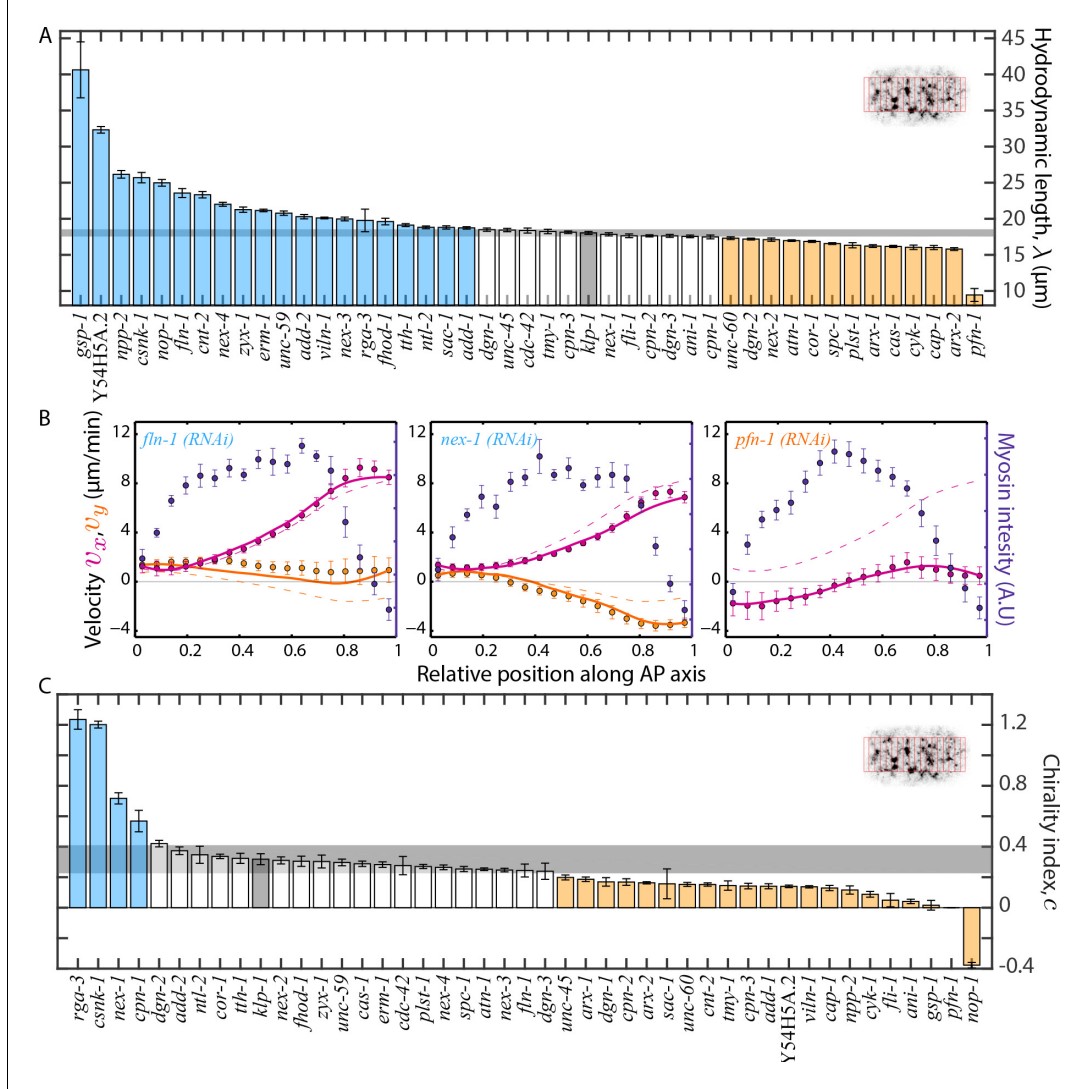

**Figure 6.** Estimation of physical properties of the cortex. (A), (C) Comparison of the hydrodynamic length, $\lambda$ and chirality index, $c$ of the cortex respectively. Error bars, SEM; gray bar, negative control, *klp-1 (RNAi)* condition; gray horizontal bar, error of the mean with 99% confidence for *klp-1 (RNAi)*; cyan, beige bars, significantly different knockdowns with 99% confidence (significance determined using normal cumulative distribution function in MATLAB). The bins over which spatial average of velocities were determined in each frame are shown in the inset. (B) Average myosin intensity (blue markers) and velocity profiles (magenta markers, AP flow velocity $v_x$; beige markers, y-velocity $v_y$) along the AP axis for representative RNAi conditions. Error bars, SEM. Magenta and beige curves, respective theoretical velocity profiles. Dashed lines, theoretical velocity profiles for *klp-1 (RNAi)* condition. See *Figure 6—figure supplement 1* for more examples of fit profiles of significantly different ABPs. See Supplementary file for number of independent embryo samples in each RNAi condition.

DOI: https://doi.org/10.7554/eLife.37677.017

The following figure supplement is available for figure 6:

**Figure supplement 1.** Comparison of hydrodynamic model fit to experimental flow profiles.
DOI: https://doi.org/10.7554/eLife.37677.018

correspond to a full loss of function, and our approach allows for a quantitative characterization of subtle cortical phenotypes. This has enabled us to identity a number of ABPs that were not previously described to have a significant impact on cortical structure and dynamics. We also identified cortex bulk physical properties (hydrodynamic length, chirality index) that are under control of ABPs and actomyosin regulators. Notably, the large-scale physical properties of the cortex are important for embryonic development wherein, a sufficiently long hydrodynamic length facilitates establishment of head-tail (AP) body axis (*Mayer et al., 2010*) and chiral active torques in the cortex help

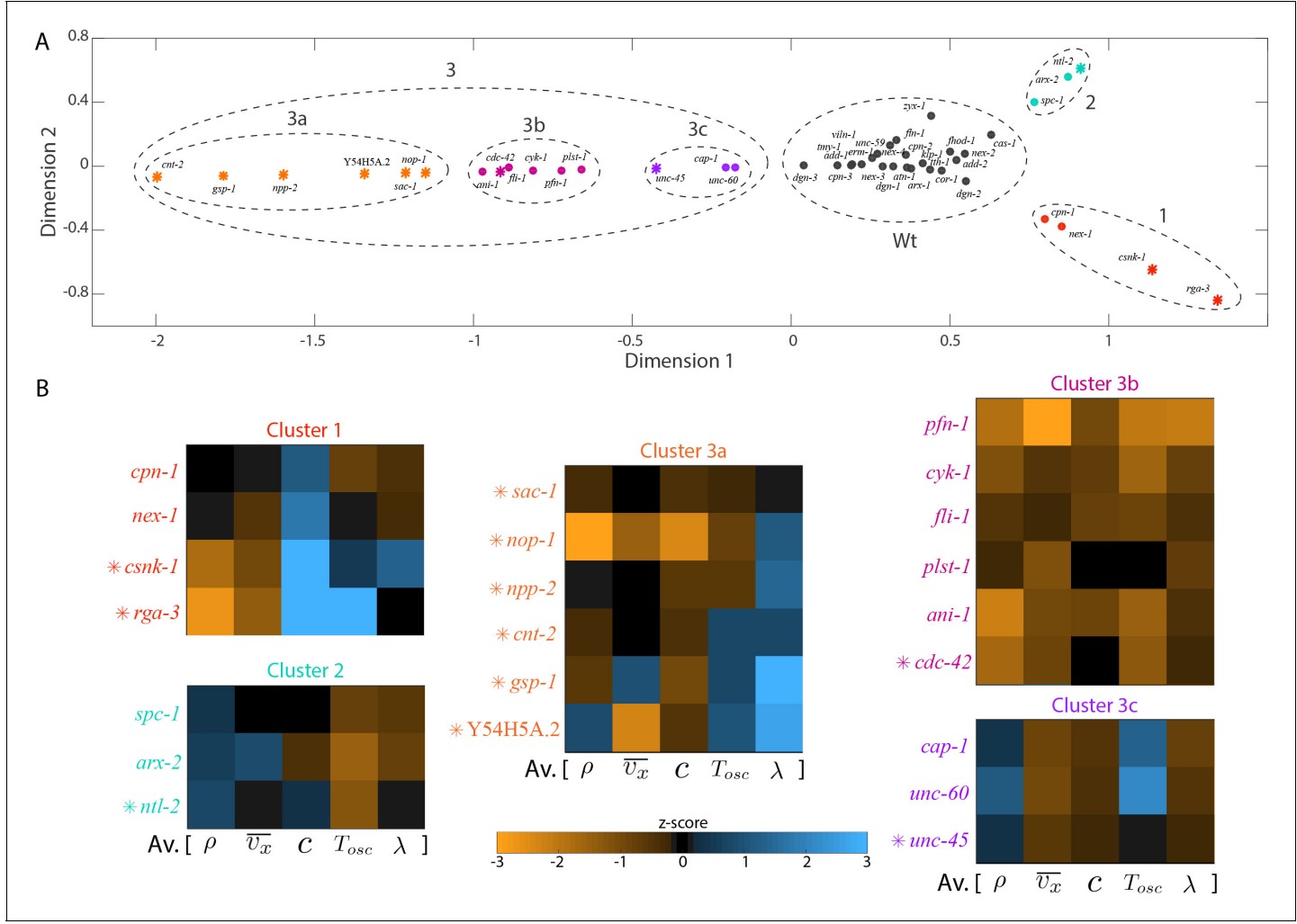

**Figure 7.** Distinct molecular functions possess similar roles at large-scales. (**A**) Non-linear dimensional reduction of the quantitative data set was performed and plotted along dimensions that represent the highest variability. Circular markers, ABPs; star markers, actomyosin regulators; colours represent the different clusters and black represents knockdowns in the same group as the negative control, *klp-1 (RNAi)*; Wt - wild type. See ***Figure 7—figure supplement 1*** for a heat map of the wild type cluster and ***Figure 7—figure supplement 4*** for a correlation of quantifications and the two principal axes of the dimensionally reduced data set. (**B**) Parameters that were used for dimensional reduction (foci density $\rho$, AP velocity $\overline{v_x}$, chirality index *c*, flow period $T_{osc}$, hydrodynamic length $\lambda$) were rearranged and displayed as heat maps for knockdowns that constitute each cluster. Colour bar, z-score. See ***Figure 7—figure supplement 2*** for correlations between different quantifications and ***Figure 7—figure supplement 3*** for visualisation of cytoplasmic actin fluorescence for *unc-60* and *cap-1 (RNAi)* from cluster 3c.

DOI: https://doi.org/10.7554/eLife.37677.019

The following figure supplements are available for figure 7:

**Figure supplement 1.** Heat map of proteins part of the wild type cluster.

DOI: https://doi.org/10.7554/eLife.37677.020

**Figure supplement 2.** Correlation between quantifications for ABPs.

DOI: https://doi.org/10.7554/eLife.37677.021

**Figure supplement 3.** Cytoplasmic actin fluorescence.

DOI: https://doi.org/10.7554/eLife.37677.022

**Figure supplement 4.** Correlation between quantifications and clustering dimensions.

DOI: https://doi.org/10.7554/eLife.37677.023

determine the L/R body axis (***Pohl and Bao, 2010***; ***Schonegg et al., 2014***; ***Singh and Pohl, 2014***; ***Naganathan et al., 2014***; ***Naganathan et al., 2016***). Since these material properties are impacted by several gene knockdowns, we suggest that the observed hydrodynamic length and chirality index (18 μm and 0.321 respectively for *klp-1 (RNAi)*) emerges from numerous molecular interactions and

are a 'summed effect' of many different protein activities. Hence, RNAi perturbation of individual molecular activities results in subtle yet quantifiable deviation from the characteristic wild type behaviour.

Chirality is an emergent active property of the actomyosin cytoskeleton that is dependent on molecular-scale torque generation (*Naganathan et al., 2014*). Here, we identify actin binding proteins to play a significant role in regulating chirality suggesting that cortical actin network architecture is equally important. This resonates well with a recent discovery, where in addition to myosin activity, the actin network architecture was shown to play a vital role in regulating cortex thickness and tension (*Chugh et al., 2017*). Finally, actomyosin pulsation is another property of the actomyosin cortex that is involved in a number of morphogenetic processes (*Gorfinkiel, 2016*). While self-organization of actomyosin cortex (*Munjal et al., 2015*; *Nishikawa et al., 2017*) and pulsation of Rho-associated proteins (*Nishikawa et al., 2017*) were suggested to be important for actomyosin pulsation, here we identify ABPs that facilitate the emergence of pulsation. Generally, we observe that a reduction in cortical viscosity is correlated with faster oscillations (cluster 3b proteins). Furthermore, spectrin (cluster 2) knockdown led to faster oscillations and given that spectrin is vital for maintenance of membrane shape (*Zhang et al., 2013*), it will be interesting to investigate the effect of membrane stability on actomyosin oscillations.

Interestingly, our MCE analysis indicates that distinct molecular activities can give rise to similar functional phenotypes at large-scales. We suggest that this is likely because the same material property or the same physical behaviour is being impacted by distinct sets of proteins from a single group. For example, polymerizing, crosslinking and bundling activities of ABPs are known to modulate viscoelastic responses of actin networks (*Stricker et al., 2010*). Here, we show that knockdown of any of these molecular activities leads to similar large-scale phenotypes (cluster 3b in *Figure 7*) and results in a decreased hydrodynamic length. This signifies a reduced cortical viscosity in all these conditions and thus likely leading to similar flow phenotypes. Similarly, reducing the severing activity or capping of actin filaments leads to similar phenotypes (cluster 3c in *Figure 7*) that signified an increase in friction of the cortex with the cytoplasm. Our results suggest that critical physical parameters that determine large-scale functions can be tuned by modulating a spectrum of molecular functions. In other words, there exists several ways in which a bulk physical property can be achieved. Our results are indicative of a 'morphogenetic degeneracy', where distinct activities at the molecular scale contribute to similar physical activities at larger scales.

Any screening approach where a small set of cellular functions and activities are probed with a large number of molecular perturbations will give rise to degeneracy. This is because there will be cases where similar outcomes in terms of changes to the cellular functions assayed will be achieved with distinct molecular perturbations, and this is most obvious for cases where there is a redundancy in molecular function. We are here suggesting that there is a mechanism at work in living systems that is distinct from this 'trivial degeneracy'. Our line of argumentation is that there is an intermediate layer at the mesoscale that is important to consider. Phenotypes at larger scales are determined by unusual material properties of living matter at this mesoscale, together with force and torque generation. Now, material properties at the mesoscale arise through coarse-graining and averaging microscopic degrees of freedom, and only a small set of mesoscopic physical parameters contribute to the force balance and the dynamics of the system at the mesoscale. This again gives rise to a degeneracy, but this 'morphogenetic degeneracy' at the mesoscale is of physical nature and can arise in the absence of redundancy of molecular activity. Since coarse-graining implies a reduction of the degrees of freedom of a system, similar large scale behaviours can emerge from distinct molecular processes that need not necessarily be redundant with each other.

To conclude, it is important to connect molecular functions to phenotypes at larger scales, and the approach we have pursued here exemplifies an experimental strategy that allows to map out this physical degeneracy between molecular activities and mesoscale physical properties that determine phenotypes. We speculate that the degeneracy we observe in the functioning of ABPs contributes to the robust performance of the actomyosin cytoskeleton. Finally, while degeneracy and its link to robustness has been widely studied with respect to gene regulatory networks (*Whitacre, 2010*; *Whitacre and Bender, 2010*; *Edelman and Gally, 2001*), we here suggest a novel physical implication of degeneracy in living systems that contributes to the robustness of biological matter.

## Materials and methods

### *C.elegans* strains

In this study, TH455 transgenic line (*unc-119(ed3) III; zuIs45[nmy-2::NMY-2::GFP + unc-119(+)] V; ddIs249[TH0566(pie1::Lifeact::mCherry:pie1)]*) was used for imaging cortical flow and SWG007 transgenic line *nmy-2(cp8[NMY-2::GFP + unc-119(+)]) I; unc-119(ed3) III; gesIs001[unc-119(ed3) III; (mex-5::Life act::mKate2::nmy2u + unc-119(+))]* for imaging cytoplasmic actin. *C. elegans* worms were cultured on OP50-seeeded NGM agar plates as described (*Brenner, 1974*).

### RNA interference

We first compiled a list of 95 ABPs by assessing putative functions (see *Supplementary file 1*) and known expression patterns. We removed from our experiments ABPs that act as enzymes, and because our analysis was restricted to 1-cell embryos, proteins that have muscle-specific or cell-cell adhesion-specific functions were also removed. The final list comprised 33 ABPs, which were subjected to RNAi-based knockdown to investigate their roles in actomyosin-based cortical flows. RNAi experiments were performed by feeding (*Timmons et al., 2001*). Early L4-staged worms were placed on feeding plates (NGM agar containing 1 mM isopropyl-$\beta$-D-thiogalactoside and 50 µgml$^{-1}$ ampicillin) and incubated for 40 hrs at $25°C$. We defined feeding time (number of hours of RNAi) as the time between transfer of worms to the feeding plate and putative fertilization of the egg. Worms were dissected in M9 buffer and the embryos were mounted on 2% agarose pads for image acquisition. The number of hours of feeding was reduced by 10 hrs if 40 hr feeding resulted in sterile worms or produced embryos that arrested at meiosis. The feeding time had to be reduced for *cap-1* (30 hrs), *cyk-1* (30 hrs), *pfn-1* (20 hrs) and *unc-60* (15 hrs) RNAi conditions.

Feeding clones for *cdc-42*, *plst-1*, *erm-1*, *unc-45*, *unc-59* and the actomyosin regulators were obtained from the Ahringer lab (Gurdon institute, Cambridge, United Kingdom), and for *pfn-1* and *unc-60* from the Hyman lab (MPI-CBG, Dresden, Germany). The rest of the feeding clones were obtained from Source Bioscience (Nottingham, United Kingdom).

### Image acquisition

Cortical flow movies using the TH455 transgenic line were acquired at $23 - 24°C$ with a spinning disc confocal microscope using a Zeiss C-Apochromat 63X/1.2 NA objective lens, a Yokogawa CSU-X1 scan head, a 525/50 nm bandpass emission filter from Semrock and an Andor iXon EMCCD camera (512 by 512 pixels). A stack consisting of three z-planes (0.5 µm apart) with a 488 nm laser (30% AOTF intensity) and an exposure of 150 ms was acquired at an interval of 5 s from the onset of cortical flow until the first cell division. The maximum intensity projection of the stack at each time point was then subjected for further analysis.

### Flow velocity analysis

Flow velocity analysis was performed from the start of flow until pseudo-cleavage. We refer to this analysis period as the 'entire flow period'. 2D cortical flow velocity fields were obtained by performing Particle Image Velocimetry (PIV) (*Raffel et al., 2007*) using the freely available PIVlab MATLAB algorithm (pivlab.blogspot.de). PIVlab was employed by performing a 3-step multi pass (with linear window deformation) where the final interrogation area was 16 pixels with a step of 8 pixels.

2D velocity fields were projected to the AP axis by dividing the embryo into 18 bins along the AP axis (see *Figure 6A* inset and Figure 1B in *Naganathan et al., 2014*), and by spatially averaging the x-component or the y-component of velocity along each bin in a single frame. Average AP velocity $\overline{v_x}$ was quantified in each frame by spatially averaging the x-component of velocity in the posterior across bins 13 to 16. The posterior domain was considered for velocity analysis because of high flow velocities (see *Figure 6B*). In the anterior domain, because the flow velocities are relatively lower (see *Figure 6B*), it is harder to distinguish changes in flow velocities across RNAi conditions. Average chiral counter rotation velocity $\overline{v_c}$ was quantified in each frame by subtracting the y-component of velocity in the anterior (spatially averaged across bins 3 to 6) from the y-component of velocity in the posterior (spatially averaged across bins 13 to 16). Histograms of $\overline{v_x}$ and $\overline{v_c}$ were plotted by computing these quantities in each frame across the entire flow period in all embryos of one experimental

condition and mean velocities were compared between conditions. Significantly different knock-downs with 99% confidence were determined using the Wilcoxon rank sum test in MATLAB.

To obtain the flow profiles, 2D velocity fields were projected to the AP axis and averaged over time across the entire flow period (from start of flow till pseudo cleavage). These time-averaged flow profiles were then averaged across all embryos for one experimental condition. Bin extent in the $y$ direction was restricted to a stripe of about 13 µm.

## Oscillation analysis

A box of size 6 µm by 6 µm was defined in the posterior with the box centre in between bins 14 and 15. Average AP velocity quantified in this box was used for determination of oscillation period. This analysis was performed in individual embryos. The temporal change in average AP velocity was determined in the box and autocorrelation of the resulting temporal profile was determined using a custom-written function. Before performing autocorrelation, the mean AP velocity in the box across the entire flow period was determined and subtracted from the data. Autocorrelation was performed by shifting incrementally the mean subtracted data until half the total analysis time. The first peak in the autocorrelation profile, which corresponds to the flow oscillation period, was determined using the 'lmax' function downloaded from Mathworks. Very few movies across conditions resulted in no peak in the autocorrelation profile, which were discarded. In the case of negative control, the first peak could be identified in all movies. A mean oscillation flow period was determined across embryos for a single experimental condition and compared. Significantly different knockdowns with 95% confidence were determined using the Wilcoxon rank sum test in MATLAB.

## Foci structure analysis

A characteristic myosin foci size was determined by performing spatial myosin fluorescence intensity autocorrelation in MATLAB. The autocorrelation was performed in a stripe of about 27 µm wide (centred in between bins 7 and 8) and 13 µm high in the anterior of the embryo (see *Figure 4B* inset). The anterior domain was considered for foci structure analysis given the higher overall signal of myosin (see *Figure 6B*). In the posterior, flow depletes myosin in addition to turnover, thus interfering with the foci structure analysis. A radial average of the resulting autocorrelation was performed using the 'rscan' algorithm downloaded from Mathworks. In this function, the coordinates of a circle with a specific radius around an origin are determined, following which the average values of points where the circle passes through are computed. The radius of the circle is incremented and a radially averaged profile is obtained. The position of the first local minimum in the autocorrelation profile, defined as the characteristic foci size, was then determined using the 'lmin' function downloaded from Mathworks.

Foci number density was determined in the same analysis stripe by a thresholding method. An interrogation area of about 2 µm by 2 µm was chosen and shifted pixel by pixel through the entire stripe. All local maxima in fluorescence intensities were first identified. The identified maxima were then taken through a number of steps to determine foci number density. Firstly, if the identified maxima were closer than 1 µm, they were merged into a single maximum. Secondly, foci that were significantly dimmer than the rest were identified and discarded. This was performed by first determining the mean fluorescence intensity of each interrogation area centred around the identified maxima. A threshold fluorescence intensity was defined by calculating the average intensity $\pm$ 0.5*STD across all interrogation areas of interest. Any interrogation area with a mean intensity lesser than the threshold was considered to be dimmer than the rest and therefore discarded. Finally, foci that comprised many local maxima were identified and merged into a single focus. This was performed by determining the fluorescence intensities in the region between identified maxima. The maxima were merged into a single focus, if no continuous decrease in intensity between the maxima for more than a micron was observed. The total number of foci thus identified was divided by the area of the stripe to obtain the number density.

Histograms of characteristic foci sizes and number densities were plotted by computing these quantities in each frame during the first 75 s of cortical flow in all embryos of one experimental condition and mean values were compared between conditions. Significantly different knockdowns with 99% confidence were determined using the Wilcoxon rank sum test in MATLAB.

## Hydrodynamic model and fitting phenomenological coefficients

The theory for cortical flows that we use to extract the hydrodynamic length and the chirality index is the result of a rigorous derivation from a generic phenomenological theory for active chiral fluids (*Fürthauer et al., 2013*). The steps of the derivation are laid out in the Appendix of (*Naganathan et al., 2014*). The resulting theory is compressible in two dimensions but is built on top of a three-dimensional incompressible fluid theory. The important assumptions in the derivation of the theory are: (i) the cortex is viscous on the time scales of cortical flows and thus has no elastic term. This is in fact a consequence of cortical turnover and one might guess that a cortex without turnover might behave more like an elastic gel. Assuming a viscous cortex therefore indirectly takes into account actin and myosin turnover. (ii) A specific relation between the bulk viscosity and shear viscosity of the 3D actomyosin gel was assumed: they are set equal. This is done for simplicity. (iii) The friction term can be considered linear and local. This model cannot describe the observed oscillatory behaviour in the actomyosin cortex. Actomyosin oscillations are driven by independent oscillations in the Rho signalling pathway that regulates myosin activity (*Nishikawa et al., 2017*) and an independent model is therefore required to describe oscillatory flows.

To compare our theory to experiment, we numerically solved hydrodynamic equations of a thin film of an active chiral fluid (*Naganathan et al., 2014*) using a finite difference scheme, using the two extreme points of the measured velocity profile as boundary conditions. These equations depend on three free parameters, the hydrodynamic length $l$, and the constants $\alpha$ and $\beta$ which relate the measured fluorescence intensity $I$ to the active tension and the active torque density such that $T = \alpha I$ and $\tau = \beta I$, respectively. Myosin is the main driver of cortical flow. This has been extensively validated for anterior-directed flows in (*Mayer et al., 2010*) and for chiral flows in (*Naganathan et al., 2014*). These papers show that active stresses are growing functions μ(C) of myosin concentration $c$. The exact functional form is unknown and might very well depend on the structure of the underlying actomyosin cortex. Here, we report the first order co-efficient (linear response co-efficient) of a Taylor expansion of μ(C) and its chiral counterpart. Note that our theory makes no predictions on how the structure of the cortex affects flows.

We adjusted these three parameters by performing a least squares fit of the solutions of these equations to the measured velocity profile. In this way, we obtained values for the hydrodynamic length $l$ and the chirality index $c = \tau/T$ numerically. We obtained error estimates for $l$, $\alpha$ and $\beta$ from the Hessian of the residual of the least square fit with respect to the parameters.

The *pfn-1 (RNAi)* phenotype was characterized by extreme and fast motion on timescales that are short (15 s) compared to our averaging window and importantly approach the cortical turnover time (30 s), see *Figure 3*. The hydrodynamic model, which we developed is strictly valid in a regime where the cortex is viscous, that is for flows which are slow compared to the cortical turnover time. *pfn-1 (RNAi)* might push the cortex into a regime where this assumption breaks down. Indeed, movies of *pfn-1 (RNAi)* showed events that might be interpreted as local ruptures of the cortex, which is an elastic effect. Since our theory does not include elastic effects, it failed to capture a phenotype dominated by elastic effects. A second effect that comes into play due to the more violent flows is that local compression can lead to local alignment of actin filaments in the cortex. This changes the symmetries of the problem and may give rise to effects that are not captured in our model, which assumes that the actin network is isotropic in the cortical plane. Active fluid theory without chirality was therefore used to fit velocity profiles of *pfn-1 (RNAi)* as the fit quality was poor when chirality was included. Therefore, we could determine the hydrodynamic length but not the chirality index for *pfn-1 (RNAi)*.

## Nonlinear dimensional reduction

Quantifications from both the ABP and suppressor screen data set (11 actomyosin regulators) were first combined and the zscore (from MATLAB) was computed for each RNAi condition across each quantification. Since chirality index for *pfn-1 (RNAi)* could not be determined from a fit of velocity profiles and given that chiral counter-rotation velocity for *pfn-1 (RNAi)* was nearly zero (see *Figure 2C*), a chirality index of zero was used for *pfn-1 (RNAi)* before computing the z-scores. This was then followed by determination of curvilinear distances (also called nonlinear sample distances) between data points by calculation of pair-wise distances over their minimal spanning tree constructed in the parameter space using the correlation norm (*Cannistraci et al., 2010*). Kruskal

method was used to build the minimum spanning tree. A matrix of these nonlinear pairwise distances (named minimum curvilinear kernel) was then subjected to singular value decomposition to identify the two principal nonlinear and orthogonal directions along which the highest variation in the data set was observed. The zscore normalized data set was then visualized along these two dimensions. All these steps form the Minimum Curvilinear Embedding (MCE) algorithm that is a type of tuning-free nonlinear principal component analysis (*Cannistraci et al., 2013*).

## Acknowledgements

We thank Michel Labouesse, Peter Gross, Masatoshi Nishikawa and Priyamvada Chugh for a critical reading of the manuscript. SWG was supported by the DFG (SPP 1782, GSC 97, GR 3271/2, GR 3271/3, GR 3271/4), the European Research Council (grant No 281903), ITN grants 281903 and 641639 from the EU, the Max-Planck-Society as a Max-Planck-Fellow, and the Human Frontier Science Program (RGP0023/2014). We thank the light microscopy facility of the BIOTEC/CRTD for excellent support. SRN (LT000078/2016) and SF (LT000871/2014) acknowledge support from the Human Frontier Science Program for their postdoctoral fellowships.

## Additional information

### Competing interests

Julie Ahringer: Reviewing editor, *eLife*. Frank Jülicher: Reviewing editor, *eLife*. The other authors declare that no competing interests exist.

### Funding

| Funder | Grant reference number | Author |
| --- | --- | --- |
| Max-Planck-Gesellschaft | Open-access funding | Sundar Ram Naganathan |
| Human Frontier Science Program | LT000078/2016 | Sundar Ram Naganathan |
| Human Frontier Science Program | LT000871/2014 | Sebastian Fürthauer |
| Human Frontier Science Program | LT00960/2006 | Josana Rodriguez |
| Deutsche Forschungsgemeinschaft | SPP 1782 | Stephan W Grill |
| European Research Council | 281903 | Stephan W Grill |
| H2020 Marie Skłodowska-Curie Actions | ITN 281903 | Stephan W Grill |
| Human Frontier Science Program | RGP0023/2014 | Stephan W Grill |
| Deutsche Forschungsgemeinschaft | GSC 97 | Stephan W Grill |
| Deutsche Forschungsgemeinschaft | GR 3271/2 | Stephan W Grill |
| Deutsche Forschungsgemeinschaft | GR 3271/3 | Stephan W Grill |
| Deutsche Forschungsgemeinschaft | GR 3271/4 | Stephan W Grill |
| H2020 Marie Skłodowska-Curie Actions | ITN 641639 | Stephan W Grill |
| Herchel Smith Postdoctoral Fellowship | | Josana Rodriguez |
| Royal Society | Research Grant RG2015R2 | Josana Rodriguez |

| Newcastle University | Faculty Fellowship | Josana Rodriguez |

The funders had no role in study design, data collection and interpretation, or the decision to submit the work for publication.

### Author contributions

Sundar Ram Naganathan, Conceptualization, Resources, Data curation, Software, Formal analysis, Validation, Investigation, Visualization, Methodology, Writing—original draft, Writing—review and editing, Designed the research, Performed the experiments and analysis; Sebastian Fürthauer, Software, Formal analysis, Validation, Investigation, Methodology, Writing—original draft, Writing—review and editing, Designed the research, Performed comparisons to hydrodynamic theory; Josana Rodriguez, Data curation, Writing—review and editing, Performed the suppressor screen; Bruno Thomas Fievet, Data curation, Performed the suppressor screen; Frank Jülicher, Supervision, Writing—review and editing, Performed comparisons to hydrodynamic theory; Julie Ahringer, Conceptualization, Supervision, Project administration, Writing—review and editing, Performed the suppressor screen; Carlo Vittorio Cannistraci, Software, Methodology, Writing—review and editing, Performed non-linear dimensional reduction; Stephan W Grill, Conceptualization, Resources, Formal analysis, Supervision, Funding acquisition, Investigation, Visualization, Methodology, Writing—original draft, Project administration, Writing—review and editing, Designed the research

### Author ORCIDs

Sundar Ram Naganathan (iD) https://orcid.org/0000-0001-5106-8687
Sebastian Fürthauer (iD) https://orcid.org/0000-0001-9581-5963
Josana Rodriguez (iD) http://orcid.org/0000-0001-8800-1283
Frank Jülicher (iD) https://orcid.org/0000-0003-4731-9185
Julie Ahringer (iD) https://orcid.org/0000-0002-7074-4051
Carlo Vittorio Cannistraci (iD) https://orcid.org/0000-0003-0100-8410

### Decision letter and Author response

Decision letter https://doi.org/10.7554/eLife.37677.027
Author response https://doi.org/10.7554/eLife.37677.028

## Additional files

### Supplementary files

• Supplementary file 1. Table of ABPs and suppressor proteins. ABPs and suppressor proteins that were tested as well as those that were discarded are compiled. Common names of the proteins, their orthologous *C. elegans* gene names, putative functions, number of hours of RNAi and the number of biological replicates performed for each knockdown are indicated. Star indicates ABPs that were also picked up in the suppressor screen previously performed in the Ahringer lab.
DOI: https://doi.org/10.7554/eLife.37677.024

• Transparent reporting form
DOI: https://doi.org/10.7554/eLife.37677.025

### Data availability

All data generated or analysed during this study are included in the manuscript and supporting files.

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
