## [Decision Letter]

Thank you for submitting your article "Morphogenetic degeneracies in the actomyosin cortex" for consideration by *eLife*. Your article has been reviewed by Naama Barkai as the Senior Editor, a Reviewing Editor and two peer reviewers. The following individual involved in review of your submission has agreed to reveal his identity: Stefan Wieser (Reviewer #1).

The reviewers have discussed the reviews with one another and the Reviewing Editor and agreed that this paper provides an important contribution to the literature. Several points in the paper, however, require further discussion, as is detailed in the individual reviewers. Please address all comments below in the revised version.

*Reviewer #1:*

S.R. Naganathan and co-authors present an RNAi based screen combined with quantitative live-imaging to identify molecular candidates underlying the large-scale physical properties of the cell cortex.

This work is a very interesting contribution towards advancing our current understanding of how individual molecular players determine distinct emergent properties at the cytoskeleton level which are related to cellular function. The authors perform a rigorous and extensive image-based RNAi interference screen in the *C. elegans* zygote to map diverse molecular actin binding proteins (ABPs) and actin regulatory proteins to bulk mechanical properties of the cell cortex such as flow speed /*v*, hydrodynamics length scale /*λ* and chirality /*c*. These large-scale actomyosin properties create a parameter space with low correlation to identify the contribution of molecular components to mostly independent large-scale properties. The authors identify previously unrecognized molecular modulators of actomyosin cortex dynamics and their analysis reveals that different molecular players converge on modulating the same large-scale cortical properties. This leads the authors to propose a "morphogenetic degeneracy" in which multiple molecular components modulate the same emergent physical cortex characteristics. They speculate that this behaviour is potentially relevant for the robustness of dynamic cellular processes in development.

This work is a relevant contribution to the field of cytoskeleton and cellular dynamics with interesting resources and data sets relevant to a broad scientific community. The article is very well structured and written and presents and summarises the results in an appropriate way.

One concern is related to the argument about "morphogenetic degeneracy". Although this argument catches attention, the finding that different molecular players impact on the same global physical properties of a composite material such as the cell cortex might be expected. The authors speculate about the potential impact of degeneracy on robustness – which is exciting – but awaits future experimental tests.

*Reviewer #2:*

This paper presents an in-depth quantitative study of the role of actin binding proteins (ABPs) in the dynamics of the actomyosin cortex of *C. elegans* zygotes. The authors perform a rather extensive RNAi screen of ABPs, associated with a quantitative analysis of actomyosin dynamics, which allows to extract several biophysical parameters (friction length, active torque, period of oscillations etc.). This leads to the quantification of the effect of various molecular players in cortical dynamics. The authors then identify classes of ABPs having a similar impact on the cortex dynamics and put forward the concept of morphogenetic degeneracy.

The paper is very well written, clear, and based on solid observations and arguments. A quantitative analysis of the effect of ABPs on actomyosin dynamics at this level is new to the best of my knowledge and will be surely useful for the community. To me the only disappointing aspect of such study is probably the lack of surprising conclusion or novel effect (which again does not diminish the value of their quantitative analysis). The authors put forward in the discussion the concept of morphogenetic degeneracy. In my opinion their approach inherently leads to some degeneracy, since it relies on a fitting procedure to a coarse-grained physical model, which by definition accounts for the zoo of microscopic degrees of freedom (ABPs) through only a few mesoscopic parameters (e.g. viscosity, friction). It is important that the authors justify why their results go beyond such a trivial degeneracy. To me the interesting side of this discussion would rather be the clustering into classes of ABPs with similar functions; the robustness of such classification however not clear at this stage and is likely to depend on the choice of hydrodynamic model, clustering algorithm.

To conclude, given the quality of the results I recommend publication, provided that the authors provide a more critical discussion of the observed degeneracy that claim. The following points should also be addressed for clarity:

– The choice of hydrodynamic model has the merit to be probably the simplest one that provides good fit to data. Since it neglects pressure like forces in the gel it does not require to explicitly take into actin turn-over dynamics. This can give the erroneous impression that the dynamics is independent of actin turn over dynamics. Since the effect of ABPs acting on actin turn-over is analysed, this should be commented. The limitations of the model should be briefly stated (newtonian fluid, linear friction among others).

– In the fitting procedure the authors take the active stress as proportional to myosin intensity. This should be commented upon. In particular the authors show that myosin clustering (size and density) can have opposite effects, which could sound contradictory.

– The hydrodynamic model as given does not lead to oscillatory behaviour, those require other ingredients. This should be stated.

– In subsection “Dimensional reduction analysis reveals clusters with similar effects on cortical mechanics”, the authors say: “Increasing F-actin density throughout the cytosol is likely to increase cytoplasmic viscosity, which in turn would lead to an increased effective friction and a reduced hydrodynamic length.” Given the definition of friction length as the ratio of viscosity and friction (root of) I don't see why.

– As a general comment, the author analysed isolated perturbations of ABPs, and therefore miss cooperativity in ABPs, which might be important.

---

## [Author Response]

Reviewer #2:[…]To conclude, given the quality of the results I recommend publication, provided that the authors provide a more critical discussion of the observed degeneracy that claim. The following points should also be addressed for clarity:

We thank the reviewer for the encouraging comments. In response to the ‘trivial degeneracy’, this is a very good point that perhaps was not clear in our original submission. We have now rewritten much of the discussion and have added a new paragraph that discusses this point. Briefly, we are suggesting that there is an intermediate layer at the mesoscale that is important to consider, and that has not been systematically investigated before. Phenotypes at larger scales are determined by unusual material properties of living matter at this mesoscale, together with force and torque generation. Now, material properties at the mesoscale arise through coarse-graining and averaging microscopic degrees of freedom, and only a small set of mesoscopic physical parameters contribute to the force balance and the dynamics of the system at the mesoscale. This gives rise to a degeneracy, but this ‘morphogenetic degeneracy’ at the mesoscale is of physical nature and can arise in the absence of redundancy of molecular activity. We think that this is an important distinction to make, and we here outline an approach that allows to map out this physical degeneracy between molecular activities and mesoscale physical properties that determine phenotypes.

Finally, we would like to note that the clustering result is not entirely dependent on the coarse-grained model we have used, and has quantifications (flow velocity, oscillation period and foci structure) that do not depend on our physical model. We have now included a new figure that shows the contribution of each quantification to the final clustering result (Figure 7—figure supplement 4).

Degeneracy is easy to understand from a physics perspective but important to investigate in biology where they are likely to provide a cell/tissue/embryo with robustness against noise and perturbations (Wagner, 2005; Whitacre, 2010; Whitacre and Bender, 2010). A key question for the future will be to investigate the extent to which degeneracy facilitates further evolution and innovation of regulatory systems. Moreover, to address the link between physical degeneracy and biological robustness in the case of cortical flow, an analysis of PAR dynamics and polarity establishment across actin binding protein knockdown conditions is essential, which is beyond the scope of this manuscript.

We have now included these arguments in the discussion, which was rewritten to a large extent in respond to this and other concerns.

– The choice of hydrodynamic model has the merit to be probably the simplest one that provides good fit to data. Since it neglects pressure like forces in the gel it does not require to explicitly take into actin turn-over dynamics. This can give the erroneous impression that the dynamics is independent of actin turn over dynamics. Since the effect of ABPs acting on actin turn-over is analysed, this should be commented. The limitations of the model should be briefly stated (newtonian fluid, linear friction among others).

The theory for cortical flows that we use to extract the hydrodynamic length and the chirality index is the result of a rigorous derivation from a generic phenomenological theory for active chiral fluids (Fuerthauer et al., 2013). The steps of the derivation are laid out in the Appendix of Naganathan et al. (2014). The resulting theory is compressible in two dimensions but is built on top of a three-dimensional incompressible fluid theory. The important assumptions in the derivation of the theory are: (i) the cortex is viscous on the time scales of cortical flows and thus has no elastic term. This is a consequence of cortical turnover and one might guess that a cortex without turnover might behave more like an elastic gel. Assuming a viscous cortex therefore indirectly takes into account actin and myosin turnover. (ii) A specific relation between the bulk viscosity and shear viscosity of the actomyosin gel was assumed, and they are set equal. (iii) The friction term can be considered linear and local.

The referee is correct that this ratio of one is a simplification. It is, however, possible to motivate different ratios of bulk to shear viscosity, depending on how the transition from a 3D bulk active chiral fluid to a 2D surface active chiral fluid is pursued. We would like to point out here that we had in the course of our original work where we presented our discovery of chiral flows in *C. elegans* (Naganathan et al., 2014) also utilized different ratios of this quantity. However, since we have decoupled flows in x- from flows in y-velocities in this earlier manuscript, changing this ratio essentially has the effect of simply renormalizing other physical parameters like the chirality index, without an apparent impact on the quality of fit. Hence, we had chosen to set the ratio to one for simplicity. In this manuscript here, we also decouple flows in x- from flows in y-direction in our analysis, and thus have kept this ratio at one as well.

We have now included a brief discussion in the Materials and methods section.

– In the fitting procedure the authors take the active stress as proportional to myosin intensity. This should be commented upon. In particular the authors show that myosin clustering (size and density) can have opposite effects, which could sound contradictory.

Myosin is the main driver of cortical flow. This has been extensively validated in Mayer et al. (2010) and for chiral flows in Naganathan et al. (2014). These papers show that active stresses are growing functions μ(*c*) of myosin concentration *c*. The exact functional form is unknown and might very well depend on the structure of the underlying actomyosin cortex. Here, we report the first order coefficient (linear response coefficient) of a Taylor expansion of μ(*c*) and its chiral counterpart. Our theory makes no predictions on how the structure of the cortex affects flows. We now better explain this in the methods section.

– The hydrodynamic model as given does not lead to oscillatory behaviour, those require other ingredients. This should be stated.

The reviewer is right in stating that the current hydrodynamic model does not lead to oscillatory behavior. Oscillations in the actomyosin cortex were shown in an earlier work (Nishikawa et al., 2017) to depend on independent oscillations in the Rho signaling pathway that regulates myosin activity. This can be incorporated into a similar hydrodynamic model, which goes beyond the scope of the current manuscript. We have now explicitly stated this in the methods section.

– In subsection “Dimensional reduction analysis reveals clusters with similar effects on cortical mechanics”, the authors say: “Increasing F-actin density throughout the cytosol is likely to increase cytoplasmic viscosity, which in turn would lead to an increased effective friction and a reduced hydrodynamic length.” Given the definition of friction length as the ratio of viscosity and friction (root of) I don't see why.

Unlike an increased F-actin density in the cortex itself, which would affect the cortical viscosity and its effective friction in the same way, thus leaving the hydrodynamic length unchanged, an increase in cytosolic F-actin will only affect the boundary condition between the cortex and the cytoplasm. Thus, it would increase the friction and not the cortical viscosity leading to a reduced hydrodynamic length.

– As a general comment, the author analysed isolated perturbations of ABPs, and therefore miss cooperativity in ABPs, which might be important.

We agree with the referee that ABPs are likely to function in a cooperative fashion and that considering the interaction between ABPs might lead to interesting and unexpected results. This however goes beyond the scope of the present paper.

References:

Wagner A. (2005) Distributed robustness versus redundancy as causes of mutational robustness. BioEssays, 27(2):176-88